Thylacinus (Marsupialia: Thylacinidae) from the Mio-Pliocene boundary and the diversity of Late Neogene thylacinids in Australia

Yates Adam M. adamm.yates@nt.gov.au
Museum and Art Gallery of the Northern Territory, Museum of Central Australia , Alice Springs, Northern Territory , Australia
Wedel Mathew
Electronic publication date: 2015 May 12
Publication date: 2015
Volume: 3
Electronic Location ID: e931
Received 2015 Mar 18; Accepted 2015 Apr 13
Copyright: © 2015 Yates
Copyright year: 2015
Copyright holder: Yates
License: This is an open access article distributed under the terms of the Creative Commons Attribution License, which permits unrestricted use, distribution, reproduction and adaptation in any medium and for any purpose provided that it is properly attributed. For attribution, the original author(s), title, publication source (PeerJ) and either DOI or URL of the article must be cited.
License URL: https://creativecommons.org/licenses/by/4.0/

Keywords: Thylacinus, Thylacinidae, Neogene, Australia, Diversity

Funding: The author received no funding for this work.

==============================
Thylacinus yorkellus is described as a new, moderately small-bodied species of thylacinid from the latest Miocene or, more likely, earliest Pliocene of South Australia. The new species can be diagnosed by the autapomorphic presence a strongly developed precingulid that terminates in a cuspidule on the anterobuccal face of the paraconid of the lower molars and a tiny basal anterior cuspidule on P2, P3 and the lower molars. It is found by cladistic analysis to be the sister species of the recently extinct Th. cynocephalus and distinct from the approximately coeval Th. megiriani from the Northern Territory. New dentary material is described and referred to Th. megiriani. These add character data and allow this species to be re-diagnosed based on autapomorphic character traits. Each of the three known late Miocene to early Pliocene Thylacinus species (Th. potens, Th. megiriani and Th. yorkellus) suggest that, instead of declining, there was a modest radiation of Thylacinus in the late Miocene.

Introduction

The tragic tale of Thylacinidae, or the marsupial wolves, involves a sustained loss of diversity from an early to middle Miocene high (Wroe, 2003) up to the final extermination of the last surviving population of Thylacinus cynocephalus at our own hands in the early twentieth century (Thornback & Jenkins, 1982; Fisher & Blomberg, 2012). Just when the Thylacinidae became restricted to a single species is an interesting question that might relate to broadscale changes in Australian terrestrial ecosystems during the latter part of the Cenozoic. It has been noted that the late Miocene diversity of thylacinids is severely depleted (Rich, 1991; Wroe & Muirhead, 1999; Wroe, 2003), although the dearth of fossil deposits from this time is a confounding factor. Nevertheless one popular account has suggested that the wolf-sized Th. potens from the Alcoota Local Fauna (Woodburne, 1967) was the sole surviving lineage by this time and that it was directly ancestral to the modern Th. cynocephalus (Archer, Hand & Godthelp, 1991).

Subsequently Tyarrpecinus rothi, a small plesiomorphic thylacinid that is separate from the genus Thylacinus, was discovered in the Alcoota Local Fauna demonstrating that at least two thylacinid lineages were surviving at this point in the Late Miocene (Murray & Megirian, 2000). However, all thylacinids from younger Local Faunas belong to Thylacinus, indicating that Ty. rothi was a late surviving relict that probably died out shortly after the deposition of the Alcoota Local Fauna. The discovery of Ty. rothi does not falsify the hypothesis that Th. potens was part of a single anagenetic lineage of large-bodied thylacinids that led directly to Th. cynocephalus. Therefore, the scenario where Thylacinidae was reduced to a single lineage prior to the close of the Miocene remained a viable hypothesis.

A further large-bodied Thylacinus species, Th. megiriani, was discovered in the Ongeva Local Fauna which derives from a channel-fill in the Waite formation that overlies the Alcoota Local Fauna on Alcoota Station (Murray, 1997). The Ongeva Local Fauna correlates with the Beaumaris Local Fauna from the Black Rock Sandstone in the Port Philip Basin of south-eastern Australia based on the shared presence of Zygomaturus gilli (Megirian, Murray & Wells, 1996; Murray, 1997; Megirian et al., 2010). The Black Rock Sandstone also preserves a marine invertebrate fauna and has a robustly stratigraphically controlled age of 6.2–5.0 ma, an age that straddles the Mio-Pliocene boundary (Dickinson et al., 2002). Th. megiriani shares more derived features with the modern Th. cynocephalus than with Th. potens. Thus Th. megiriani could be interpreted as chronospecies on an anagenetic lineage leading from Th. potens to Th. cynocephalus. At the time of its description, Th. megiriani was known only from its holotype,a single broken maxilla with worn and damaged teeth. However, two large thylacinid dentary fragments have been found in the Ongeva Local Fauna since the initial publication that can be referred to this species. Character data from these specimens has been incorporated into a phylogenetic analysis of thylacinid relationships (Yates, 2014) but they have not been described or adequately illustrated in the scientific literature.

There is a further thylacinid from the penecontemporaneous Curramulka Local Fauna of South Australia (Pledge, 1992) that sheds light on the number of thylacinid lineages present in Australia during the latest Miocene and earliest Pliocene. This thylacinid is described here as a new species. In addition, the new mandibular specimens of Th. megiriani are described and the species diagnosed with autapomorphic characters. This new data casts strong doubt on the hypothesis of a single anagenetic lineage of Thylacinus in the late Neogene of Australia.

Systematic Palaeontology

Dasyuromorphia Gill, 1872	
Thylacinidae Bonaparte, 1838	
Thylacinus Temminck, 1824	
Thylacinus yorkellus sp. nov.	
urn:lsid:zoobank.org:act:D176F98E-740B-4E37-A491-3AFDCCC1CD50	

Holotype. South Australian Museum (hereafter SAM) P29807, incomplete left dentary with C, P1−3 and M2−3 (Figs. 1–2 and Table 1).

Figure 1 Thylacinus yorkellus, holotype SAM P29807, incomplete left dentary.

Photographs of the specimen. (A) Lateral view (B) medial view (C) dorsal view. Scale bar = 50 mm. Photographs by Steven Jackson.

Figure 2 Thylacinus yorkellus holotype SAM P29807, incomplete left dentary.

Interpretive drawings of the specimen. (A) lateral view (B) medial view (C) dorsal view. Abbreviations: adl, anterior cuspidule; abdl, anterobuccal cuspidule; amf, anterior mental foramen; c, canine; cn, carnassial notch; end, entoconid; hld, hypoconulid; hyd, hypoconid; ia, incisor alveoli; m1-3, molars 1–3; p1-3, premolars 1–3; pad, paraconid; pcd, precingulid; pmf, posterior mental foramen; prd, protoconid; sym, symphyseal surface. Hatched areas represent broken bone surfaces, grey areas represent wear facets on the teeth. Scale bar = 50 mm.

Table 1 Dental measurements of Thylacinus yorkellus.

A ∼ symbol indicates that the measurement is approximate and is taken from the alveolus.

	H (mm)	L (mm)	W (mm)	trig W (mm)	tal W (mm)	
SAM P29807						
C	15.2	8.7	6.5	–	–	
P 1	–	7.2	3.1	–	–	
P 2	–	9.8	4.0	–	–	
P 3	–	10.4	4.4	–	–	
M 1	–	∼8.7	–	–	–	
M 2	–	11.0	–	5.5	12.0	
M 3	–	12.0	–	6.5	5.7	
SAM P38799						
M 3	–	11.3	–	6.0	6.0	
Notes.

H Height of the crown, from crown-root junction to the apex

L Anteroposterior length of the crown (note that in the canine this measurement is taken at the base of the crown)

W Maximum buccolingual width of the crown

trigW buccolingual width of the trigonid

talW buccolingual width of the talonid

Referred specimen. SAM P38799, crown of right M3 (Figs. 3–4 and Table 1).

Figure 3 Thylacinus yorkellus, paratype SAM P38799, right M3.

(A) Photograph in buccal view. (B) Photograph in lingual view. (C) Interpretive drawing of A. (D) Interpretive drawing of B. Abbreviations: abdl, anterobuccal cuspidule; adl, anterior cuspidule; cn, carnassial notch; end, entoconid; hld, hypoconulid; hyd, hypoconid; pad, paraconid; pcd, precingulid; prd, protoconid. Scale bar = 10 mm. Photographs by Steven Jackson.

Figure 4 Thylacinus yorkellus, paratype SAM P38799, right M3.

(A) photograph in anterior view. (B) photograph in posterior view. (C) photograph in occlusal view (D) Interpretive drawing of A. (E) Interpretive drawing of B (F) Interpretive drawing of C. Abbreviations: abdl, anterobuccal cuspidule; adl, anterior cuspidule; cdo, cristid obliqua; cn, carnassial notch; end, entoconid; hld, hypoconulid; hyd, hypoconid; pacd, paracristid; pad, paraconid; pcd, precingulid; ppacd, preparacristid; prcd, protocristid; prd, protoconid. Scale bar = 10 mm. Photographs by Steven Jackson.

Locality and stratigraphic age. Corra-Lynn Cave, approximately 3 km south of Curramulka, York Peninsula, South Australia. Late Miocene or, more likely, early Pliocene in age.

Etymology. From York Peninsula and the diminutive suffix–ellus (Latin), referring to its small size relative to Th. cynocephalus.

Diagnosis. Th. yorkellus differs from all other thylacinids by the presence of the following autapomorphies. The lower molars have a strongly developed precingulid that terminates in a cuspidule on the anterobuccal face of the paraconid and a tiny basal anterior cuspidule on P2, P3 and the lower molars. Aside from these two autapomorphies, it can be further distinguished from Th. macknessi and all thylacinids not in the genus Thylacinus by the complete absence of metaconids on M2, M3 and, presumably, M4. It can be further distinguished from Th. potens by its smaller size (estimated adult body mass of 16–18 kg vs. 39–56 kg for Th. potens; Yates, 2014), by the presence of wide diastemata between each of the lower premolars as well between P3 and M1, and by the presence of deep cleft-like carnassial notches on the lower molars. It can be further distinguished from Th. megiriani by its smaller size (estimated body mass of Th. megiriani is 57 kg; Wroe, 2001), wider diastemata separating P3 from M1 and P2 from P3, its relatively gracile and buccolingually compressed anterior dentary lacking a ventrolateral torus, the complete absence of any trace of the metaconid on the lower molars, and by the presence of deep cleft-like carnassial notches on the lower molars. Lastly, it can be further distinguished from Th. cynocephalus by the lengths of both P2 and P3 exceeding that of M1 and by the presence of a diastema between the canine and P1.

Remarks. Pledge (1992) figured and briefly described the holotype specimen in his description of the Curramulka Local Fauna. He suggested that it might belong to an undescribed species but declined to name it. With our improved knowledge of thylacinid diversity and the addition of a second specimen showing the same autapomorphic characters of the molars as the holotype, there is now sufficient evidence to warrant the naming of a new species.

Description

The holotype includes the incisor alveoli, the canine, all three premolars, M2, M3 and the alveolus for M1 (Figs. 1–2 and Table 1). Posteriorly the dentary has broken off at the level of the posterior end of M3, so that M4 is missing. The dentary is transversely compressed, as it is in Th. cynocephalus but unlike the transversely broad, robust dentaries of Th. megiriani and some Th. potens. The dentary of Th. yorkellus differs further from that of Th. megiriani and Th. potens in lacking a thickened torus along the ventrobuccal margin, below the mental foramina. The dentary depth below the anterior root of M2 (the posteriormost level at which this measurement can be made) is 24.5 mm which lies within the range of Th. cynocephalus. Given that the length of the lower molar row and the heights of the individual molars lie below the range seen in Th. cynocephalus, I propose that the holotype Th. yorkellus has a deeper jaw relative to tooth size in comparison to the former species. A relatively deep dentary is also seen in some, but not all, specimens of Th. potens (Yates, 2014). The buccal surface of the dentary is pierced by three mental foramina set approximately at the mid height of the dentary, although it is possible that a fourth was present in what is now a large area of missing bone below P3 (Fig. 2). The anterior mental foramen is a large anteriorly facing opening with a diameter of 3.2 mm located below the anterior root of P2. The two close-set posterior mental foramina are placed below M2 and the anterior margin of M3, respectively. These openings are smaller, with anteroposterior diameters of 2.1 and 1.4 mm, respectively. In buccal view the alveolar margin is concave with P3 set lower than the rest of the preserved tooth row. The symphyseal surface extends posteriorly to the level of the anterior margin of P3.

The canine has a procumbent root and a vertically projecting crown. The tip is gently recurved so that the anterior margin in lateral view is convex and the posterior margin is concave. In anterior view the crown has a weak lingual curvature. The cross-section of the crown is a slightly buccolingually compressed oval. A weakly developed carina extends along the anterolingual margin of the crown, from base to apex. The posterior surface is rounded and smooth. The measurements of the canine are given in Table 1.

A diastema of 3.3 mm separates the canine alveolus from that of P1. P1 is a smaller, lower-crowned tooth than all of the other postcanine teeth. In occlusal view it is elongately ovoid, with the long axis oriented anteroposteriorly. Although the tip of the protoconid is well worn, it is clear that the unworn tooth would have had a height less than its anterior-posterior length. The apex of the protocone is located in the anterior half with an elongate heel extending posterior to it. The heel has a sharp posterior corner which forms an incipient cuspidule. In buccal view the posterior margin of the protoconid curves gently down to the posterior heel. A weakly developed cristid extends along the anterior margin of the protoconid. The tooth has no other cristids or cuspidules.

A diastema of 2.8 mm sepparates P1 from P2. P2 differs from P1 in being larger (Table 1) and proportionately taller. The apex of the triangular protoconid is less skewed to the anterior side of the tooth, being placed slightly mesial to the mid length. A weak cristid extend up the anterior side of the protoconid. There is a minute tubercle-like cuspidule developed at the base of this anterior cristid. A posterior cristid extends from the apex of the protoconid to the posterior corner of the basal heel. A distinct inflection in the buccal profile marks the change from the heel to the posterior margin of the protoconid.

A long diastema of 6.4 mm sepparates P2 from P3. P3 is larger than, P2 and differs from it in a number of subtle features. P3 shows a greater development of the anterior basal cuspidule than in P2. The large protoconid is centrally located and triangular in buccal view. Both anterior and posterior margins bear cristids. In occlusal view the crown is widest posteriorly, where the heel forms an incipient talonid basin. A shallow notch in the buccal profile separates the posterior heel from the protoconid, and the posterior heel forms a protrusive cuspidule with a worn tip.

M1 is missing, and the lingual alveolar margin of the dentary is missing from the level of the posterior root of P3, posteriorly. Nevertheless, the buccal side of the M1 alveolus is separated from P3 by a diastema of 3.0 mm, whereas there is no diastema between the alveolus of M1 and M2.

M2 consists of two large trigonid cusps (paraconid and protoconid) and three far smaller cusps (hypoconid, hypoconulid and entoconid) surrounding the talonid basin. The most remarkable feature of M2 (and M3) is the strength of the precingulid, which forms a sharp-edged shelf that terminates in a well-developed cuspidule on the anterobuccal face of the base of the paraconid. The precingulid slopes steeply posteroventrally from this cuspidule and terminates near the base of the crown level with the anterior margin of the protoconid. The paraconid is well developed and triangular in occlusal view. A minute cuspidule is developed on the anterior slope of the paraconid, near the base of the crown, at about the same level as the anterobuccal cuspidule of the precingulid. The protoconid forms the largest cusp of M2. In buccal view the anterior margin of the protoconid curves posteriorly towards the apex, however this shape has been created by wear. A distinct, tear-drop shaped wear facet is developed at the tip of the protoconid and extends down its anterior side, along the course of the paracristid. A deep carnassial notch is developed between the paraconid and protoconid, dividing the paracristid into two sections. There is no trace of a protocristid or a metaconid on the posterior side of the protoconid, although the cristid obliqua extends approximately half way up this surface. The cristid obliqua is also divided into two sections by a deep carnassial notch, this time separating the protoconid from the hypoconid. In buccal and lingual view the talonid is set distinctly lower than the trigonid. In occlusal view its buccolingual width is slightly greater than that of the trigonid. All of the cusps of the talonid are highly worn and flat-topped. The largest of these and the most strongly projecting is the buccally placed hypoconid. The posteriorly located hypoconulid has been virtually obliterated by wear and is only recognisable as a slight thickening of the enamel around the posterior rim of the wear facet. The entoconid is placed on the lingual side of the talonid, slightly inset from the lingual margin of the crown. A short ridge extends posterobuccally from its apex towards the hypoconulid. A postcingulid extends obliquely up the posterior surface of the hypoconid from a basally located point on the buccal side to a more apical point on the lingual side.

M3 is very much like M2, differing in the following respects. The minute anterior cuspidule at the base of the paraconid is less well-developed. The protoconid is taller, and the cristid obliqua extends for less than half its height up the posterior surface. The talonid is slightly narrower than the trigonid and the hypoconulid forms a low conical cuspid. The posterobuccal ridge extending from the apex of the entoconid joins the anterior base of the hypoconulid, thus creating an oval basin that is separated from, and much smaller than, the main talonid basin. A low but sharp postcristid between the hypoconulid and the entoconid forms the posterolingual margin of this oval basin. The postcingulid is very reduced, and is little more than a shallow depression on the posterior surface of the hypoconid.

The M3 of SAM P38799 shows some minor differences from the holotype but the main features are the same, including the distinctive precingulid terminating in an anterobuccal cuspidule (Figs. 3–4 and Table 1). In this specimen, the anterior cuspidule at the base of the paraconid is better developed than in either of the two molars of the holotype, and has an incipient preparacristid leading from it towards the apex of the paraconid. The talonid is as wide as the trigonid and the entoconid appears to lack the posterobuccal ridge that links it to the hypoconulid.

Body mass estimate

Calculating from the third lower molar length regression formula, derived from the dasyuromorphian-only data set of Myers (2001), a body mass estimate of 17.8 kg is obtained for the holotype and 15.9 kg for SAM P38799. These estimates are well below the average body mass of 29.5 kg for recent Tasmanian Th. cynocephalus (Paddle, 2000) but are not outside the range estimated for mainland Holocene samples of this species (Letnic, Fillios & Crowther, 2012). However, it is clear that Th. yorkellus was a far smaller species than either Th. potens or Th. megiriani which have estimated body masses ranging between 38.7 and 57.3 kg (Wroe, 2001; Yates, 2014).

Thylacinus megiriani (Murray, 1997)

Holotype. Museum and Art Gallery of the Northern Territory (hereafter designated NTM) P9618, fragmentary left maxilla with P1−3 and M1−3

Referred specimens. NTM P4376, anterior fragment of a right dentary containing the alveoli with broken roots for P2, P3 and M1 and the empty alveolus for M2 (Figs. 5–7). NTM P4377, posterior fragment of a right dentary containing an incomplete M4 (Figs. 8–10).

Figure 5 Thylacinus megiriani, NTM P4376, fragmentary right dentary.

Monochrome photographs of the specimen after whitening with ammonium chloride. (A) Lateral view (B) medial view (C) dorsal view. Scale bar = 50 mm. Photographs by Steven Jackson.

Figure 6 Thylacinus megiriani, NTM P4376, fragmentary right dentary.

Colour photographs of the specimen. (A) Lateral view (B) medial view (C) dorsal view. Scale bar = 50 mm. Photographs by Steven Jackson.

Figure 7 Thylacinus megiriani, NTM P4376, fragmentary right dentary.

Interpretive drawings of the specimen. (A) Lateral view (B) medial view (C) dorsal view. Abbreviations: amf, anterior mental foramen; m 1–2, molars 1–2; mf, mental foramen; p 1–2, premolars 1–2; pmf, posterior mental foramen; sym, symphyseal surface; vt, ventral torus. Hatched areas represent broken bone surfaces, grey areas represent patches of matrix. Scale bar = 50 mm.

Figure 8 Thylacinus megiriani, NTMP4377, fragment of right dentary including partial right M4.

(A-C) monochrome photographs of the specimen after whitening with ammonium chloride. (D-F) interpretive drawings of (A)–(C). (A, D) lateral view. (B, E) medial view. (C, F) dorsal view. Abbreviations: al, alveolus for anterior root of m4; cdo, cristid obliqua; cn, carnassial notch; cp, base of coronoid process; ed, entoconid; mcd, metacristid; md, metaconid; pcd, postcristid; prcd, protocristid; prd, protoconid. Grey areas represent patches of matrix, hatched areas represent broken bone or tooth surfaces. Scale bar = 30 mm. Photographs by Steven Jackson.

Figure 9 Thylacinus megiriani, NTMP4377, fragment of right dentary including partial right M4.

Colour photographs of the specimen. (A) Lateral view (B) medial view. Scale bar = 30 mm. Photographs by Steven Jackson.

Figure 10 Thylacinus megiriani, NTM P4377, right M4 in posterolingual view.

(A) photograph. (B) Interpretive drawing. Abbreviations: cdo, cristid obliqua; hyd, hypoconid; mcd, metacristid; md, metaconid; prcd, protocristid; prd, protoconid. Grey areas represent patches of adherent matrix. Scale bar = 5 mm. Photograph by the author.

Locality and stratigraphic age. ‘South Quarry,’ an excavation on the south-western side of ‘Hill 1’ (Woodburne, 1967) an erosional remnant of the upper Waite Formation on the Alcoota Fossil Reserve. Ongeva Local Fauna, latest Miocene or earliest Pliocene in age.

Diagnosis. Th. megiriani differs from all other thylacinids by the presence of the following two autapomorphies. There is a a short, lobe-like postcingulum between the metastyle and protocone of M2. A hypertrophied torus is developed along the ventrobuccal margin of the dentary so that its buccolingual width at the level of P3 is greater than 75% of its depth at the same level. The presence of a small stylar cusp E on M2 and M3 might also be an autapomorphic reversal within the genus Thylacinus but the optimisation of the character is ambiguous due to its presence in the closely related outgroup, Wabulacinus ridei, and the absence of a preserved M2 or M3 for Th. macknessi. Similarly, the presence of a vestigial metaconid and metacristid on M4 (and presumably the other lower molars) is ambiguous due to the presence of metaconids in Th. macknessi and their absence in Th. potens. This may represent an autapomorphic reversal in Th. megiriani or a case of convergent loss in Th. potens and the Th. yorkellus + Th. cynocephalus clade. Th. megiriani can be further distinguished from Th. macknessi and all non-Thylacinus thylacinids by: its greater size; the great reduction of the size of the paracone relative to the metacone; elongation of the postmetacrista, so that it is greater than 52% the length of the tooth in M2 and M3. It can be further distinguished from Th. potens by: the long axis of P1 aligned with those of the other upper premolars; the length of M1 is greater than its width; the complete absence of a precingulum on M1 and M3; the absence of a metaconule on all upper molars; M3 much longer (>5%) than M2; M3 longer than wide; presence of a diastema between P3 and M1.

Remarks. The new dentary specimens cannot be referred to Th. megiriani on the basis of autapomorphic characteristics due to their fragmentary nature and lack of overlap with the holotype maxilla. However, the apparent reversal to the presence of an, albeit vestigial, metaconid in NTM P4377 parallels the reappearance of a tiny tubercular stylar cusp E in the upper molars of the holotype, indicating that the two features may be not be independent in thylacinids and provides further support for the referral of the dentary fragment NTM P4377 to Th. megiriani. It is interesting to note that the expression of the metaconid and stylar cusp E consistently covary in all known thylacinids, ie. taxa either possess both cusps or neither, never one or the other.

Furthermore, the new dentary fragments derive from the same level in the same quarry as the holotype, are of an appropriately large size, and can be distinguished from Th. potens, which is the only other large-bodied thylacinid known from a similar time and place. For these reasons it is reasonable to refer them to Th. megiriani.

Description of the new material

The anterior dentary fragment (NTM P4376; Figs. 5–7, Table 2) has only broken roots and empty alveoli and does not contain any tooth crowns. This coupled with poor preservation means that it is not immediately obvious exactly which teeth the roots and alveoli belong to and therefore the disposition of the teeth in the jaw. However, several lines of evidence indicate that the posterior two empty alveoli are for the anterior and posterior roots of M2, respectively, while the posterior two projecting root stumps are the anterior and posterior roots of M1. Anterior to the molar alveoli there are the broken roots and alveoli of P2 and P3 (Fig. 7). Evidence for this interpretation comes from a partial bridge of finished bone between the second and third preserved alveoli, indicating that there was a diastema between two different teeth in this position, rather than these alveoli representing the anterior and posterior alveoli of the same tooth. The posterior limit of symphyseal surface is level with the anterior margin of the fourth alveolus. As the posterior end of the symphysis draws level with P3, or at least the diastema between P2 and P3 in all thylacinid specimens examined, the third and fourth alveoli in NTM P4376 should represent the anterior and posterior alveoli for P3. From the alveoli it can be seen that there was no diastema between M1 and M2, but there was a short diastema of 3.6 mm between P3 and M1, and a slightly longer diastema of 4.0 mm between P2 and P3.

Table 2 Dental measurements of Thylacinus megiriani.

A ∼ symbol indicates that the measurement is approximate and is taken from the alveolus. Measurements in brackets are taken as preserved and represent minimum values due to incompleteness of the specimen.

	H (mm)	L (mm)	trigW (mm)	talW (mm)	
NTM P4376					
P 2	–	∼15.6	–	–	
P 3	–	∼18.0	–	–	
M 1	–	∼14.3	–	–	
M 2	–	∼13.4	–	–	
NTM P4377					
M 4	16.9	–	(7.3)	6.5	
Notes.

H Height of crown, from crown-root junction to tip of protoconid

L Basal length

trigW buccolingual width of trigonid

talW buccolingual width of talonid

The dentary is dorsoventrally shallow but buccolingually thickened, giving it a distinctly robust appearance. Adding to the thickness of the jaw is a strong torus developed along the ventrobuccal margin of the dentary from a point level with the midlength of P2 to one level with the distal end of M1. At the maximum thickness of the torus (level with the posterior root of P3) the dentary measures 20.8 mm wide, which approaches the dorsoventral depth of the dentary at the same level (27.0 mm). The buccal surface of the dentary, above the ventrobuccal torus, is pierced by three mental foramina. The anterior-most mental foramen lies anterior to the alveoli for P2, the middle foramen lies level with the anterior end of P3, and the posterior foramen lies level with the posterior end of M1.

The second specimen (NTM P4377; Figs. 8–10 and Table 2) is a small fragment from the posterior end of a right dentary including a partial M4 and the anterior base of the coronoid process. The lingual surface of the dentary is gently convex dorsoventrally, whereas the buccal surface bears a ridge that slopes posterodorsally to merge with the leading margin of the coronoid process. The buccal ridge forms the lower border of a flat dorsobuccally facing surface that occurs below the fourth lower molar. The ventral part of the dentary is missing so the depth to height ratio cannot be measured. The incomplete M4 includes the talonid and the posterior side of the protoconid. The protoconid is far taller than the talonid with an almost vertically oriented posterior margin. The protocristid extends from the apical tip of protoconid to a tubercle about half-way down the posterior face of the protoconid (Fig. 10). This tiny tubercle appears to be a vestigial remnant of the metaconid which is usually absent in derived species of Thylacinus. Two weakly developed cristids extend ventrally from the tubercle. The two cristids diverge at a highly acute angle. The more buccally placed cristid is the cristid obliqua. It extends to the base of the protoconid and up the anterior face of the hypoconid. A shallow, weakly developed carnassial notch is created between the hypoconid and the protoconid. This notch is similar to the weak notches developed on the lower molars of Th. potens and unlike the deep clefts that can be observed in the carnassial notches of the lower molars of Th. yorkellus and Th. cynocephalus. The lingual cristid that branches from the tubercle on the posterior face of the protoconid is interpreted as a metacristid. It terminates at the base of the protoconid without extending onto the talonid. Its presence strengthens the interpretation of the tubercle as a vestigial metaconid. The moderately large, conical hypoconid comprises the talonid which bears no other cusps. The anterobuccal face of the hypoconid is planed off by an oblique wear facet that faces buccodorsally. The weak cristid obliqua extends directly anteriorly from the apex of the hypoconid. The anterolingual rim of the hypoconid is rounded, lacking a postcristid.

Cladistic Analysis

Th. yorkellus was included in an earlier cladistic analysis of thylacinid relationships (Yates, 2014). This analysis already included data from the mandibular specimens of Th. megiriani described in this paper. The same characters from the previous analysis were employed with one modification to character 36. Previously, this character simply described the absence (0) or presence (1) of a carnassial notch in the cristid obliqua of the lower molars. In the present analysis, the derived state is divided into two states: a weakly-developed shallow carnassial notch (1) or a deep, strongly-developed carnassial notch (2). The multistate character is treated as ordered. The score for character 34 (metaconid size in m2−4) of Thylacinus megiriani was changed from 2 (completely absent) to 1 (present as a minute cuspidule) in the light of the new evidence presented above.

The resulting matrix (Appendix S1) was subjected to a maximum parsimony analysis in PAUP 4.0b (Swofford, 2002) using the following settings: heuristic search; random addition sequence with 500 replicates; and TBR branch-swapping algorithm. The strength of the internal nodes was tested with a bootstrap analysis (1,000 bootstrap replicates, heuristic searching with 50 addition sequence replicates).

The search resulted in two equally most parsimonious trees of 93 steps. The topologies of these two trees are completely congruent with the two trees obtained in the earlier analysis (Yates, 2014) with the loss of resolution entirely the result of a variable position of Maximucinus muirheadae. If this taxon is pruned a posteriori, a fully resolved tree is obtained (Fig. 11A). Th. yorkellus is found to be the sister species of the modern Th. cynocephalus. The Th. yorkellus + Th. cynocephalus clade is relatively robust with the second highest bootstrap score of all the internal clades (71%, Fig. 11B). Nevertheless, missing data reduces the number of unambiguous apomorphies of this clade to one: the presence of a deep, well-developed carnassial notch in the cristid obliqua on each of the lower molars.

Figure 11 Results of the cladistic analysis of thylacinid interrelationships.

(A) Reduced cladistic consensus tree of two most parsimonious trees (tree length = 93 steps) obtained after a posteriori pruning of Maximucinus muirheadae. (B) Strict consensus with bootstrap support values for clades with values >50%.

Discussion

The recognition of Th. yorkellus increases the number of Thylacinus species in the late Miocene—earliest Pliocene interval to three: Th. potens, Th. megiriani and Th. yorkellus. A fourth lineage, that of the recent Th. cynocephalus, can also be inferred to have arisen in this period (Fig. 12). None of the local faunas that produced the three known species can be directly dated but they do appear to form a stratigraphic sequence. The Ongeva Local Fauna occurs in a channel incised into sediments that overlie those that contain the Alcoota Local Fauna, directly demonstrating that the former is younger than the latter (Megirian, Murray & Wells, 1996). However, the absolute age difference between them is probably not great because many species are shared between the two deposits including the biostratigraphically important Kolopsis torus (Megirian, Murray & Wells, 1996). The presence of Zygomaturus gilli in the Ongeva Local Fauna indicates that it may be correlated with the Beaumaris Local Fauna from the Black Rock Sandstone of Port Phillip Bay (Megirian, Murray & Wells, 1996). The Beamaris Local Fauna straddles the Mio-Pliocene boundary on the basis of local stratigraphic controls and its included marine invertebrate fauna (Dickinson et al., 2002). The difference in age between the Ongeva Local Fauna and the Curramulka Local Fauna, or even their relative positions with respect to each other, is difficult to determine. Pledge estimated that the deposit was late Miocene in age which would make it roughly contemporaneous with, or possibly older than the Ongeva Local Fauna. However a constrained seriation analysis clustered the Curramulka Local Fauna with younger Pliocene local faunas, that were given the age-name Tirarian while the Ongeva Local Fauna was clustered with the Beaumaris and Alcoota Local Fauna in the Waitean Age (Megirian et al., 2010). A younger age for the Curramulka Local Fauna is supported by the presence of a number of macropodid genera that are not known from the older Waitean fauna, including: Baringa, Troposodon and Protemnodon (Pledge, 1992). Nevertheless, the Curramulka Local Fauna is probably one of the oldest Tirarian assemblages due to its total lack of rodent remains, despite intensive fine sieving of the silty matrix and presence of numerous other small vertebrates (Pledge, 1992). Thus, an early Pliocene age of 4.5 ma or older would appear likely based on the timing of the appearance of rodents in other fossil deposits in Australia (Breed & Ford, 2007).

Figure 12 A phylogenetic tree of Thylacinus, calibrated to the geological timescale.

Note that the dates of Th. potens, Th. megiriani and Th. yorkellus are not tightly constrained and the ages given here are approximations with the grey bars indicating a range of plausible ages. Numbers represent ages in ma. Dates of boundaries taken from Cohen et al. (2013).

These three deposits do seem to form a stratigraphic sequence, and the sequence matches the branching order of the species of Thylacinus recovered in the cladistic analysis, supporting the hypothesis of an anagentic lineage. However, the time difference between them is probably too small for a plausible anagenetic sequence. It is quite likely that all three local faunas span no more than 2 or 3 ma, from 5 ma to 7 or 8 Ma. If Th. potens, Th. megiriani and Th. yorkellus did indeed form an anagenetic lineage then the turnover of morphologically distinct forms would appear to be unrealistically fast, especially given the known range of Th. cynocephalus is nearly four million years (based on the late Pliocene age of the Chinchilla Local Fauna). Furthermore, each of these species can be diagnosed with autapomorphies, suggesting that none is an ancestor of any other.

Thus it would appear that for a brief period in the latest Miocene through to the earliest Pliocene the genus Thylacinus experienced a modest evolutionary radiation (Fig. 12). However, the products of this radiation were short-lived and by the late Pliocene there is no indication of any surviving thylacinid species other than Th. cynocephalus (Dawson, 1982). Unfortunately the Pliocene record of Thylacinus is fragmentary and sparse (Mackness et al., 2002). Louys & Price (in press) report two specimens in the Queensland Museum that have been identified as Th. rostralis (=Th. cynocephalus) and genuinely derive from the Chinchilla Local Fauna. Apart from these specimens, and Th. yorkellus, all other Pliocene occurences of Thylacinus are specifically indeterminate (Mackness et al., 2002).

Supplemental Information

Appendix S1 Character-Taxon Matrix

The character list is provided in Yates (2014).

Click here for additional data file.

I thank Ben McHenry of the South Australian Museum for access to the thylacinid specimens in his care. I also thank Mary-Anne Binnie, also from the South Australian Museum, for organising a loan of material from that institution. Eric Scott and an and an anonymous reviewer improved the manuscript with their helpful comments. All photographs used in this paper were taken by Steven Jackson, except Fig. 10 which was taken by the author.

Additional Information and Declarations

Competing Interests

Author Contributions

New Species Registration

The author is an employee of the Museum and Art Gallery of the Northern Territory.

Adam M. Yates conceived and designed the experiments, performed the experiments, analyzed the data, contributed reagents/materials/analysis tools, wrote the paper, prepared figures and/or tables, reviewed drafts of the paper.

The following information was supplied regarding the registration of a newly described species:

ZooBank:urn:lsid:zoobank.org:act:D176F98E-740B-4E37-A491-3AFDCCC1CD50.

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
