# Peer review of "Thylacinus (Marsupialia: Thylacinidae) from the Mio-Pliocene boundary and the diversity of Late Neogene thylacinids in Australia"

_PeerJ, doi:10.7717/peerj.931_

## Round 0.1 · original submission · Minor Revisions

· Academic Editor

Minor Revisions

Both reviewers found the manuscript to be scientifically sound, and their comments almost exclusively deal with relatively minor issues of grammar and word choice. I have reviewed all of their suggestions, which seem both reasonable and straightforward to implement.

Reviewer 1 ·

Basic reporting

The text of this article is generally acceptable, although there are numerous minor typos and grammatical errors throughout that need to be corrected - my comments and suggestions on this manuscript are generally cosmetic in nature. Standardization in capitalization needs to be addressed in the writing, particularly with respect to geochronological and biochronological units (e.g., the author uses both "late Miocene" [line 17] and "Late Miocene" [line 24], as well as "Alcoota Local Fauna" [ine 19-20] and "Alcoota local fauna [line 23] and "Late Miocene - Earliest Pliocene" [lines 273-274]). This should be corrected before publication. In line 34, I believe that the "sandstone" in "Black Rock sandstone" should be capitalized, as this is a formal unit. "table 1" [line 105] should be capitalized. "Sepparates" is misspelled in line 113. Line 121: "medial" should be "mesial" in this sentence. These are some of the more glaring errors in basic reporting that need to be addressed. In addition, I think that Tables 1 and 2 should be combined into a single table (as they refer to dental measurements from the same species) to save space, as should Tables 3 and 4. Headings within the combined tables could differentiate the measurements described.

The following comments reflect my own professional opinions in formatting, but are suggestions for the author and editor to consider. I generally think that the use of ambiguous terms such as "large" and "small" are uninformative to the reader in anatomical descriptions unless they are accompanied by a relative modifier, such as "larger than" or "smaller relative to" (etc.) - in the Description section, the use of the terms "small" and "large" are common and I suggest they either be removed or clarified (either through proportional or quantitative additions). Example: line 106 - "P1 is a small, low-crowned tooth." This sentence is just as informative in terms of morphological description if the term "small" is removed. I would suggest trying to remove these ambiguous terms or conform to sentences such as line 113 ("P2 differs from P1 in being larger and proportionately taller." The latter example is much more informative. In addition, the use of the term "transverse" (line 143) is not anatomically informative by itself. I would suggest being more specific by using terms such as "mediolateral width" or "buccolingual width" in this section (such terms are used in the description of Thylacinus megiriani. Also, in the caption of Figure 10, this should be "distolingual" instead of "posterolingual," in keeping with the author's previous use of mesial and distal when referring to dental anatomy of thalacynid teeth.

Experimental design

No comments. I have no problems with the methods the author has used to describe this material.

Validity of the findings

The section entitled "Size Estimates" should either be changed to "Body Mass Estimates" or other measures of "size" should be included. It might be informative to the reader to provide estimates of body (or skeletal) length in addition to body mass - the latter can (and is) influenced by many variables within a species, whereas the dimensions of the skeleton are more conservative.

Additional comments

Great job on providing a thorough, informative article on this poorly understood and represented marsupial lineage!

·

Basic reporting

In terms of content, the paper was well presented and well argued, with sufficient support (given the usual vagaries of the fossil record) for the conclusions advanced. There were a number of grammatical challenges, however; these are noted on the appended pdf.

Experimental design

The manuscript meets PeerJ's scope. The fossils under discussion are well-described, and the interpretations are well thought out and have adequate support.

Validity of the findings

The manuscript presents a thorough description of the material, then builds a convincing interpretation of the significance of the remains. The fossils all have museum catalogue numbers and so are assumed to be in discipline-specific repositories, although I do not recall seeing this explicitly stated. Speculation is minimal, and noted and explained where used.

---

## Round 0.2 · accepted · Accept

· Academic Editor

Accept

Thank you for your attention to the reviewers' suggestions and for taking them in the constructive spirit that they were offered. I'm happy to accept the revised manuscript for publication in PeerJ. I'm going to let the 'emotive' opening sentence stand.

Although it is of course up to you whether to publish the review history alongside your paper, I think that doing so would add value to the paper and to science in general. It would be nice to point to in future discussions about the utility of peer review - and in teaching students - as an example of a manuscript that was already scientifically solid getting further tightened up by going through the peer-review process.